# Position: Agentic AI Research Should Prioritize Improving Semantic Quality

## Abstract

Agentic AI systems achieve high benchmark scores yet remain constrained to simple workflows with human oversight when deployed in production. Current benchmarks measure task completion but ignore quality dimensions, which leads to unreliable and brittle workflows. Our position is that agentic AI research should prioritize improving semantic quality: ensuring that natural language programs encoding agent logic are semantically correct and not just syntactically valid. Our use case analysis indicates that critical failure modes arise from semantic inconsistencies and drift across workflows. To support our position, we analyze open research challenges and derive quality requirements specialized to agentic AI, collectively called semantic quality. Then, we design a system architecture from a generic agentic workflow, showing core functional components. We argue that programs that encode agent logic in natural language must address reliability, robustness, and observability at the level of semantic correctness beyond syntactic correctness. This paper calls for action to meet standard quality requirements by focusing research on improving the semantic quality of agent programs, which we call semantic programs. We also propose initial directions for developing reliable and robust agentic AI systems.

## 1. Introduction

**Position: Current benchmarks for agentic AI measure completion of the tasks, but ignore the quality of the system. In particular, semantic errors across multi-stage workflows involving agent reasoning and tool execution have not been sufficiently addressed, and the lack of semantic quality blocks deployment of agentic AI sys-**

**tems at scale in real-world domains. Agentic AI research must prioritize improving the semantic quality. Especially, we must improve the reliability and robustness of the natural language programs that encode agent logics. We should not dismiss this as a software engineering issue. Competing for benchmark scores without semantic quality metrics will not advance agentic AI systems.**

Agentic AI systems are rapidly transitioning from research prototypes to deployment in real-world applications (Guo et al., 2024; Wang et al., 2024; Luo et al., 2025; Pan et al., 2025). We define agentic AI systems as a large language model (LLM) centered software architecture that aims to achieve autonomous and goal-directed behavior. The current shift from generative AI to agentic AI requires capabilities that extend beyond plausible text generation, integrate into non-trivial workflows, execute natural language instructions, and eventually improve productivity (Liu et al., 2024a; Bousetouane, 2025; Schneider, 2025).

Since the core capability of agentic AI systems relies on LLMs, various failure modes caused by LLMs heavily impact the entire system. Limited reproducibility in machine learning research as well as the inconsistency of large language model generations are well-known issues (Semmelrock et al., 2025; Wang & Wang, 2025). Pan et al. (2025) reports that deployed systems significantly constrain the autonomy of agents, and most workflows are limited to a small number of steps that follow predefined patterns. More importantly, most systems rely on human oversight due to reliability concerns. In larger-scale enterprise systems, where automated workflows connect business data and agents operate across the entire life-cycle of the organization, the gap between hype and practice will be further pronounced.

In this paper, we argue that the major technical problem of the current agentic AI systems is the lack of assurance of system quality. We begin by analyzing canonical use cases and identifying potential failure modes. To understand the gaps, we analyze the open challenges in agentic AI systems and present quality requirements and the functional architecture. We argue that a new semantic quality is crucial for reliable and robust agentic AI systems, and introduce the following five attributes: **semantic contracts**, **semantic lineage**, **evidence-supported claims**, **longitudinal invariance**, and **collective reliability**. These attributes govern

[1]Anonymous Institution, Anonymous City, Anonymous Region, Anonymous Country. Correspondence to: Anonymous Author <anon.email@domain.com>.

Preliminary work. Under review by the International Conference on Machine Learning (ICML). Do not distribute.

meaning, not syntax. Therefore, it is not possible to address semantic quality from traditional software engineering practices. In the functional architecture, which reflects existing agentic AI frameworks, we confirm that all features are concentrated in the agent framework layer responsible for the runtime execution of agents over computing platforms, and the agentic AI programming layer, which is middleware that orchestrates agent workloads. This motivates the need for a new **semantic program layer**[1] on top of the other two layers to address the semantic quality of the agent logic written in **natural language programs** (Miller, 1981).

In Section 2, we present three canonical use cases and common failure modes using standard quality vocabulary (ISO/IEC/IEEE 24765, 2017). Section 3 presents the system analysis results supporting that improving the semantic quality of the semantic program is the core research challenge in agentic AI systems. Sections 4 and 5 follow with a call to action and discussions on alternative views.

## 2. Failure Modes

We review the general quality dimensions in the standard glossary. Then, we identify failure modes specific to agentic AI systems and elaborate them into semantic quality attributes.

### 2.1. System Quality Dimensions

We select three major quality dimensions that are crucial to agentic AI research by simplifying extensive software system quality models in the standards (ISO/IEC 25010, 2011). **Reliability** is defined as the ability of a system to perform required functions under stated conditions for a specified period of time. **Robustness** is the ability of a system to continue to function correctly in the presence of invalid inputs or stressful environmental conditions. **Observability** is the degree to which the internal states of a system can be inferred from its external outputs.

### 2.2. Enterprise Workflow Agents

**Scenario.** A user submits a ticket requesting a refund of the last invoice because it was billed twice. Then, the agent executes the following steps: (1) retrieve the relevant refund policy and contract terms, (2) check entitlement and compliance constraints, (3) call billing APIs to identify candidate invoices and validate a duplicate charge, (4) execute the refund upon approval, (5) update downstream systems while emitting audit logs for compliance.

**Failures.** Agentic AI could fail in the following ways. (1)

---

[1]This separation resembles the three-tier architecture (Valente, 2024) because contemporary LLM services are built around cloud computing platforms.

The ambiguity in natural language is not detected, and the agent guesses the meaning when the intent of the user is underspecified. (2) The agent generated a valid API invocation through the schema constraints, e.g., valid argument types passed to the billing API, but the value of some arguments binds to the wrong customer. Passing only syntactic validation masks critical semantic errors, leading to the propagation of errors. (3) The agent may yield an incomplete transaction, e.g., the agent successfully retrieved the refund policy and issued the refund, but failed to update the internal record. In the absence of explicit transactional semantics, the system may not reliably recover to a consistent state. (4) The agent may retrieve outdated or incomplete policy documents, leading to compliance failures.

**Take away.** The failures above often appear as valid outputs at the interface level, but they violate end-to-end correctness. We restate these failures as violations of **semantic contracts** and **semantic lineage**, where meaning does not persist or is not observable through the workflows

### 2.3. Software Engieering Agents

**Scenario.** Let us consider a representative request, "migrate service X to a new logging library by updating its usage". The software engineering agents (Liu et al., 2024a; Wang et al., 2025; Dong et al., 2025) would typically follow the following steps: (1) inspect repository structure and identify relevant modules, (2) perform multi-file edits to update dependencies and call sites, (3) run build and test in a sandboxed environment, (4) iterate a repair loop using compiler and test feedback, and (5) open a pull request containing the patch, an explanation linked to the edits, and test results.

**Failures.** Software engineering agents expose a set of failure modes rooted in the violation of semantic invariances and the lack of evidence-supported reasoning. (1) The agent didn't understand the right scope of the code dependency, modified multiple files and broke interface contracts. (2) The patch compiles locally but fails in integration because there exists a version difference in the sandbox and integration environments. (3) The agent alternates between incompatible repairs that break each other, entering an unbounded iteration of repair loops. (4) The scope of the change is so broad that the patch became unreviewable.

**Take away.** The failure highlights the importance of maintaining semantic invariance on the code execution assumptions and contexts. The oscillation of local repairs and the integration failures are the situations that violate the implicit semantic invariances across the phases in the workflow, which we call **longitudinal invariance**. Tracing the changes proposed by the agent should be linked to the test cases or the code location that causes the bug, possibly maintaining the dependency chain in the code. We restate this type of failure as missing **evidence-supported claims** in

the generation.

## 2.4. Multi-Agent Reasoning for Scientific Discovery

**Scenario.** Consider a scientific discovery use case where a researcher requests a multi-agent system to generate and verify hypotheses from collected data (Gridach et al., 2025; Majumder et al., 2025; Sun et al., 2025). The system would generate a multi-agent workflow in the following steps: (1) The master agent decomposes the given task into literature retrieval, hypothesis generation, verification, and experiments. (2) Multiple workers execute decomposed tasks in parallel or in sequence through coordination. (3) Verification agent checks the generated hypothesis with available evidence. (4) The system iterates until the master agent decides to finish running subtasks and synthesize the results from subtasks.

**Failures.** Multi-agent systems exhibit failures during coordination and emergent failure modes. (1) Agents had different interpretations that led to non-convergent or early-terminated discussions. (2) Minor changes in the coordination yield different system behavior and outcomes. (3) Some worker agent hallucinates a fact, and other agents incorporate it, but the verifier also fails to reject it. (4) The master agent waits for a worker agent that doesn't return the response, or the slowest agent dominates the overall latency. (5) As the number of agents increases, the number of messages can explode, and the coordination overhead overwhelms the cost for solving the task.

**Take away.** The failures in the multi-agent reasoning reveal that they are not merely coordination complexity issues, but they are rooted in the semantic quality across the agents. The deadlock of early termination could still happen when agents don't share the same meaning space. If the overall system behavior drastically changes due to the changes in the topology or order in agent communication, or model configurations, we cannot trust the system. We call it a **collective reliability** issue by emphasizing that the reliability must be considered in multiple entities in the flow.

## 2.5. System Requirements

Taken together, the three canonical use cases show consistent patterns that can fail in agentic AI systems. Agentic AI failure modes are rarely isolated to a single bad generation, but they arise from end-to-end breakdowns in controlling the meaning of natural language instructions and generations during multi-stage LLM inference calls or tool executions. Therefore, boundary-level validation is not sufficient to prevent the aforementioned failure modes. In enterprise workflow agents, the violation of the *semantic contract* undermines the **reliability** of the whole system. Failure to track *semantic lineage* in the workflow will not ensure the end-to-end **observability**. Recall that a collection of seman-

tically wrong generations along the workflow still passes the syntactic reliability test and provides full observability. In software engineering workflow agents, tracing the reasons behind the code edits backed up by *evidence-supported claims* is essential for avoiding failures and building trust through improved **reliability** and **observability**. In complex workflows, there are implicit constraints or invariances that must be kept throughout the phases. The **robustness** against surface form changes and maintaining *longitudinal invariants* also turned out to be an important attribute. In multi-agent reasoning workflows, interactions among agents can break down when their reliability is compromised under collective behavior. The agents should maintain **reliability** by preventing hallucination cascades and **robustness** to minor differences in the meaning spaces, coordination topology, or configurations.

Examining deeper, the failure modes share the concerns derived from LLMs. (1) Failures of maintaining **semantic consistency** across the multi-stage pipelines, namely, across task decomposition, tool invocations, etc. (2) Undesired **semantic drift** under changes in the system configurations, such as selection of LLMs or hyperparameters, interaction protocols, prompt templates, etc. These patterns indicate that traditional software-quality assurance models are not sufficient to close the gap, and they demand more research in natural language programs, going beyond benchmarking-driven research. In the next section, we formalize semantic quality requirements for agentic AI systems.

## 3. System Analysis

We analyze proposed open research challenges in the literature as high-level requirements for agentic AI systems. The literature ranges from the transition from the classical agent concept to generative and agentic AI (Schneider, 2025; Sapkota et al., 2026), practical considerations in the enterprise deployments (Luo et al., 2025; Bousetouane, 2025; Pan et al., 2025), reasoning aspects around LLMs (Wu et al., 2025), multi-agent architecture with LLMs (Guo et al., 2024; Miehling et al., 2025), the protocols (Yang et al., 2025), feedback mechanisms (Liu et al., 2025b), optimization (Du et al., 2025), security, and safety (Yu et al., 2025). We also analyze canonical use cases (Wang et al., 2025; Dong et al., 2025; Liu et al., 2024a; Shi et al., 2025; Majumder et al., 2025; Gridach et al., 2025) and the system design proposals (Dao et al., 2025; Shang et al., 2025) and evaluations (Liu et al., 2024b; Mohammadi et al., 2025).

### 3.1. Semantic Quality Requirements

The failure modes in Section 2 show that the standard notion of reliability, robustness, and observability needs to be extended in agentic AI systems. We analyzed the open challenges using a generic agentic workflow and derived the

| | Semantic contracts | Longitudinal invariance | Semantic lineage | Collective-reliability | Evidence-supports |
|---|---|---|---|---|---|
| **Reliability**
Does the system complete the jobs correctly and repeatably? | Intent/Task mapping
Domain specialized fitness
Plan completeness
Semantic tool correctness | Operational determinism
Shared state invariance
Constraint retention | Slot value normalization
Long-horizon coherence | Logical/causal consistency
Hallucination control
Runtime-orchestrated recovery | Grounded claims
Verification/reflection
Knowledge freshness
Hallucination control |
| **Robustness**
Does system degrade safely under bad inputs, external drifts? | Natural language ambiguity
Paraphrase invariance
API/Schema tolerance | Long-term adaptation
Perturbation resilience
API/Schema tolerance
Robust under model updates | Context truncation resilience
Memory corruption resilience | Coordination failure
Protocol robustness
Emergent behavior
Role drift detection | Transfer calibration
Fail-closed under outages |
| **Observability**
Can we trace the internal states, decisions and information flow in the system? | Context integrity
Semantic incident taxonomy | Replay runs
Provenance stamp per config | End-to-end semantic lineage
Meaning-aware diffs
Explainability/faithfulness
Toolchain introspection | Information flow
Bias propagation and amplification | Explainability/faithfulness
Context integrity
Code provenance
Auditing and accountability |

| Generic Agentic Workflow | ① Ingress | ② Decomposition | ③ Coordination | ④ Execution | ⑤ Synthesize | ⑥ Recovery | ⑦ Handoff |
|---|---|---|---|---|---|---|---|
| | Understand Intention
Identify Tasks | Decompose tasks
Assign subgoals | Schedule workloads
Agent/Tool Protocols | LLM inference, Tools
Multi-agents/Model | Combine generations,
verifiable constraints | Resolve conflicts,
Regenerations | Provenance per run,
replay, compliance |

*Figure 1.* Semantic Quality Requirements. Figure shows five semantic quality attributes in columns. We map each attribute to three standard quality dimensions and assign relevant system requirements. For example, semantic contracts related to reliable intent/task mapping, domain-specific fitness, plan completeness, and semantic tool correctness. The full list of requirements and definitions appears in Appendix A. Darker shading indicates which quality dimension is closely associated with each semantic quality attribute. For example, semantic contracts is closely associated with reliability.

system requirements. A generic agentic workflow is initiated by understanding the intention and identifying relevant tasks. Then, it follows the phases for task decomposition, LLM inference or tool execution, synthesizing multiple generations, recovering the observed errors or resolving conflicts, and finally returning the desired output. Figure 1 summarizes the five semantic quality attributes and shows their relation to the standard quality dimensions. Each attribute spans over three dimensions, and we assign system requirements to each dimension and attribute. Appendix A shows the full list of the requirements.

Agentic AI systems need quality assurance at the level of meaning by checking **semantic contracts**, going beyond correctness of the schema and interfaces. Since agents consume and generate natural language text, the generation should be backed up by **semantic lineage** and **evidence-supported claims**. In addition, the behavior must be consistent across runs and phases, **longitudinal invariance** and multiple agents and components, **collective reliability**. Next, we define each semantic quality attribute.

**Semantic Contracts.** *Semantic contracts* refer to the meaning-level negotiations that govern how instructions, constraints, and domain-specific rules should be interpreted and preserved through the agentic workflow. The contracts should ensure the intended semantics of the tasks, a correct set of entities and slots to refer to, the goals to achieve through a plan, and conditions and context must hold. The most common cases that violate the contracts are generating

syntactically valid output with semantic errors, incorrect entity linking, dropping the implicit conditions or constraints over the workflow, misinterpreted domain rules, or failure to meet the pre/postconditions after inference or execution.

**Semantic Lineage.** *Semantic lineage* refers to the end-to-end traceability of meaning as it flows through the agentic workflow. While semantic contracts specify the correct and intended semantics that must be agreed on, the semantic lineage ensures that those semantics are observable and explainable. Violations could occur when the system cannot provide an explanation for why a certain action was performed or track the difference between the intended outcome that complies with the postcondition and the actual outcome. When context drifts away during the workflow due to the truncation or corruption of the memory, the syntactic-level log may not guarantee the observability of the states of the whole system.

**Evidence-Supported Claims.** *Evidence-supported claims* refer to the criteria that every generation or outcome must be grounded in explicit and verifiable evidence. The source of the evidence could be fresh external knowledge or formally verifiable tools. However, such external sources are limited in general. We expect that the evidence is contained in the semantic lineage when the semantic contracts establish what semantic entity or type entails from one to the other. Therefore, the major source of the evidence is the entities, slots, and constraints that participate in the semantic contracts. As a whole, evidence-supported claims ensure

reliability through verifiable sources and observability by supplying the semantic linkages in the workflow. The violation is obviously observed when the system fails to generate semantic evidence for any single generation or outcome. In the absence of evidence, downstream phases may accumulate errors, and it may eventually compromise the reliability of the entire workflow.

**Longitudinal Invariance.** *Longitudinal invariance* refers to robust behavior that the system exhibits stable and deterministic outcomes across multiple runs under various perturbations, yet retaining the semantic invariance. The longitudinal invariance looks for the consistency across multiple executions and also semantic invariance across multiple phases. The violation could be observed when small changes in the system lead to radically different behavior. For example, re-statement of the same task or instructions leads to different plans and goals, drift in the inference calls leads to execution of different tools, and updating the infrastructure results in a significant drop in performance. It is not sufficient to pass a single run of the workflow, but multiple runs or multiple phases within a single run must be replayable if the semantics of the intention remain the same.

**Collective Reliability.** *Collective reliability* refers to reliable and robust behavior in multi-agent or multi-component agentic AI systems to maintain a stable semantics across all participants. While longitudinal invariance ensures the robustness across repeated runs or multiple phases in a single run, collective reliability concerns the coordinated behavior that the final outcome must aggregate individual outcomes. We say collective reliability is violated when agents diverge in the interpretation of common entities, a factual generation of an agent hallucinating other agents, the synthesized outcome greatly depends on the topology of the agent coordinations, etc. Even if each agent is locally reliable, the collective behavior could be semantically incorrect, yielding an unreliable system as a whole.

### 3.2. System Functional Architecture

In support of our position, we analyze system requirements and existing frameworks to derive a generic functional architecture. Figure 2 shows the three layers consisting of Agent Framework Layer (AFL), the Agentic AI Programming Layer (AAPL), and the Semantic Program Layer (SPL). Our study on existing software libraries and frameworks concludes that current research and industry best practices are all concentrated in the lower AFL and AAPL, and it is essential for the community to pursue research and development effort to ensure semantic quality of agentic AI systems by building SPL on top.

**AFL: Runtime Infrastructure.** AFL faces computing platforms and is responsible for the runtime execution of agentic AI workloads. The main tasks in this layer are serving LLMs, executing tools in a sandboxed environment, transaction quality control, and telemetry. At this level, common concerns are syntactic correctness, scalability, performance, fault isolation, and logging. Most vendor frameworks and software libraries are mapped to this layer, and we consider the features at this layer to be genuine software engineering issues.

**AAPL: Middleware for Orchestration.** AAPL is a middleware that has been addressed well in terms of research and software development. Currently, structured decoding, guardrails, state graphs, tool-calling protocols, prompt optimization, etc., are commodities for building agentic AI systems. The quality metrics are measured at the level of syntactic correctness or the schema level. The scalability and performance for running heavy workloads on top of AFL are also major concerns. Many open-source frameworks belong to this layer, and ongoing research efforts aim to improve scalability and performance.

**SPL: New Design Space.** All contemporary LLM-based agents embed natural language prompts and eventually pass textual input to auto-regressive LLMs to generate the desired text output. Concerns around the semantics of the agent program are independent of scalability or formal correctness because the root lies in the nature of natural language.

### 3.3. Semantic Programming as Natural Language Programming.

We view semantic programming in agentic AI systems as a realization of the long standing vision of natural language programming in AI research (Woods, 1973; Miller, 1981). Natural language interface to database systems (Androutsopoulos et al., 1995) echoes back in natural language to query language agents (Shi et al., 2025), understanding natural language software requirements and specifications to generate code is exactly what the software engineering agents do. In practice, vibe coding (Karpathy, 2025; Sarkar & Drosos, 2025) is rapidly shifting the programming paradigm, and it implements agentic AI systems that write code through natural language interfaces and understanding. Unfortunately, the main engines that run sophisticated coding agents are proprietary.

**SPL vs. AAPL.** In Section 2 and Section 3.1, we argue that critical and unsolved issues can be consolidated in terms of semantic quality. SPL brings them into the center and views the instructions, natural language reasoning, and plans, constraints, or domain-specific decision rules as components of the natural language program. The correctness of such programs no longer depends on the syntactic level but depends on preserving the meanings and ensuring the semantic quality across the workflows. Here, we distinguish semantic programming from agentic AI programming for the following reasons. While AAPL focuses on workflow orchestra-

| Semantic Program (SPL) | Knowledge repr. | Constraints Proc. | Quality Policy | DSL and Ontology | Logic patterns |
|---|---|---|---|---|---|
| Agent semantics and roles
Workflow constraints
Task and workflow policy
Domain/API rules
Re-usable agent logic | Declare agent profiles, typed task representation, structured/typed knowledge in frames | Logical, semantic invariance, pre post conditions, decomposition constraints | Declare policies for quality attributes, task and workflow precedence rules, conflict resolutions | Domain specific ontology, API semantic contracts, Tool usages | Re-usable agentic program libraries, common operators, business logics |
| **Agentic AI Program (AAPL)** | **Enforce contracts** | **Orchestration** | **Feedback/Repair** | **Tool governance** | **Workflow patterns** |
| Enforce semantic types
Coordinate agents
Run feedback/repair loops
Tool selection/execution
Workflow patterns | Validate structured outputs from inference calls or tool executions at schema level | Schedule inference calls, tool executions, coordination of agent interactions | Collect feedback from verifiable sources and manage repair loop on failure workloads | Manage tool usage rules, ensure schema/syntactic validation of tool usage | Re-usable and performance aware workload executions via Map/Reduce, State graphs, etc |
| **Agent Frameworks (AFL)** | **Telemetry** | **Runtime execution** | **Transport control** | **Tool sandbox** | **Protocol** |
| End-to-end logs, traces
Deterministic run manifest
QoS, rate control
Safe tool execution
Comm. interaction protocols | Structured logs, traces, metrics over entire workflow. Provide diagnostic for human oversight | Deterministic replays, package environments, run manifest, reproducible runs | High quality message exchange, handling transport queues, latency control | Safe tool execution through code sandboxes, control resource limits, isolate side-effects | Protocol adaptation, version control, standardize interactions, interoperability |

*Figure 2.* Three-layer functional architecture. The semantic program layer consolidates functional components that are relevant for writing semantically correct agent AI program logic. The agentic AI program layer supports agentic AI workflow execution and enhances workflow scalability. The agent frameworks offer the software stacks for high-quality runtime for agentic AI systems.

tion and improving the scalability and performance of the system, SPL concerns who to write high-quality semantic programs that encode agent logic in natural language to satisfy the five semantic quality attributes.

**SPL Components.** Our system design identifies five functional components in SPL.

- **Semantic Knowledge Representation**: Semantic programs should provide knowledge representation for agent profiles, task representation, and structured knowledge in frames. These frames create the *semantic contracts*.

- **Semantic Constraint Processing**: Semantic program should process constraints from logical or semantic invariance, the causal structure of actions in terms of preconditions and postconditions, and additional task-specific constraints. Semantic constraints should be propagated correctly throughout the workflow to ensure *longitudinal invariance* and provide *semantic lineage*.

- **Semantic Quality Policies**: Semantic quality attributes are encoded in the policy logic to handle the rules around workflow executions and resolving the conflicting intentions or outcomes. The policy should ensure all generations are *evidence-supported claims*.

- **DSLs and Ontology**: Semantic programs should incorporate domain-specific languages and ontology, and conform to the implicit *semantic contracts* or tool usage requirements.

- **Logic Patterns**: Semantic program should provide reusable patterns for common operators that run on the

semantic level, or codified business logics in natural language.

Collectively, the above SPL components are the basic functional features of semantic programming to realize the long-standing vision of natural language programming in modern technology. We believe that improving semantic quality through a new design space in SPL clearly contrasts with current agentic AI research and existing frameworks, in which natural language prompts remain opaque text blocks without semantic structure, semantic invariants, or traceable lineage.

### 3.4. Existing Agents and Frameworks

Next, we review the software libraries and platforms for building agentic AI systems.

**Open-source libraries.** Open-source libraries offer many of the capabilities for AAPL. For example, validating the schema through structured decoding, enforcing the type system on the generated text, content filtering with LLM-as-judge, fact checking, and controlling hallucination with guardrails (LangChain, 2025a; Pydantic, 2025a). Some libraries show strength on specific components, such as workflow construction with state-graph (LangChain, 2025b), agent orchestration (CrewAIInc, 2025), and prompt optimizations for improved task performance (Khattab et al., 2024). Although the open-source libraries are widely adopted and enable agentic AI systems development, they are all limited to the AAPL boundary and do not provide SPL features and ensure semantic quality.

**Vendor frameworks.** Vendor frameworks directly face the computing platforms that providers offer, and therefore, they provide runtime management, telemetry, and quality of service at AFL (Google Cloud, 2026; Microsoft, 2025; OpenAI, 2025; NVIDIA, 2025). In addition, vendor frameworks also adopt or allow open-source AAPL libraries to provide AAPL capabilities. The convergence across vendor frameworks and the wide adoption of open-source agentic AI libraries imply that the state of technology has matured at AFL and AAPL.

**Academic Research Baseline Agents.** On the other hand, many academic research papers focus on proposing new benchmarks and evaluating multiple baseline agents. However, they rarely address the quality of the agentic AI systems. Most, if not all, agent prototypes in research implement the agentic AI system by writing prompts as raw strings that are directly embedded in the code. This results in non-reproducibility of generation as well as evaluation because many of the evaluation systems now use LLMs.

**Take Away.** Existing software systems at AFL and AAPL have matured, at least from an AI/ML research viewpoint, leaving room only for further improvement in traditional software quality. Continuing efforts to optimize computing infrastructure and the sophistication of middleware will not solve the core failure modes of agentic AI systems. As we can see from the failure modes in canonical use cases, requirements, and the functional architecture, improving the semantic quality of semantic programs would be the direct path toward successful real-world deployment of agentic AI systems at scale.

### 3.5. Emerging Research Frameworks

We see emerging research prototypes that start to address semantic quality, due to the nature of application domains and the design philosophy of the frameworks.

**Semantic Query.** Applying LLMs in database platforms is an active area of research, and semantic query operators are trying to bring the power of natural language reasoning into structured query processing (Patel et al., 2025; Liu et al., 2025a). The semantic operators are reusable program libraries that commonly appear in data workflows, such as filtering database content with natural language queries. Although the semantic qualities are not primary quality metrics in semantic query operator systems in favor of improved running time and accuracy, we expect that future research will address the semantic quality more seriously.

**Structured Dataflow.** Another agentic AI Research framework (Gliozzo et al., 2025) narrows the application domain to large-scale structured dataflow tasks and explicitly addresses the semantic contract. Built on the idea of enforcing LLMs to operate with schema in Python types (Pydantic,

2025b), it views the agentic workflow as a structured data flow that transforms typed objects using LLMs. Such transformations are formalized as a logical transduction algebra, in which a logical transduction is defined as a stateless negotiation of meaning between agents that addresses the semantic contract between them. The framework provides reusable logic patterns that can be executed on asynchronous MapReduce programming models (Dean & Ghemawat, 2008).

## 4. Call to Action

Improving the semantic quality of agentic AI systems requires steering research priorities and system development strategies. Sections 2 and 3 show that dominant research approaches, namely, creating new benchmarks and pushing the techniques in AAPL, will not solve the core failure modes, which are rooted in the meaning-level errors. Next, we outline future directions.

4.1 **Improve Software Quality in Implementations**. Before addressing semantic quality, agentic AI systems should first meet the basic quality expectations of reliability, robustness, and observability. Many research prototypes implement LLM-based agents using raw-string prompts and ad-hoc scripts, resulting in non-reproducible and non-diagnosable behaviors. These deficiencies hinder the study of semantic quality because they make it difficult to determine whether a failure is due to a semantic error or merely from noise. Although improved code quality will not solve the semantic failures identified in this paper, it will provide the foundational stability for rigorous evaluation.

4.2 **Framework and Research for SPL**. Our system analysis shows that existing frameworks and research efforts are all concentrated on AFL and AAPL, and they lack the core mechanisms to improve semantic quality attributes. Therefore, we need new frameworks dedicated to SPL, where the agent logics are treated as natural language programs, and the semantic quality attributes serve as the primary objective and acceptance criteria. While detailed implementation could vary across designs and applications, SPL frameworks should provide programming abstractions for the following: (1) semantic knowledge representation, (2) constraint processing, (3) semantic quality management, and (4) domain-specific rules. These functional components have been individually well-studied in the past, such as formal methods, automated reasoning, semantic web technology, and model-driven engineering. The research challenge is to repurpose and unify prior research into a coherent technology for natural language programming.

4.3 **Prioritize Uncertainty Quantification for Semantic Decision-Making**. Agentic AI systems make a series of consequential decisions that propagate information

across multi-step workflows. There is an urgent need to expand upon ongoing uncertainty quantification research and pursue principled mechanisms to quantify uncertainty in semantic reasoning: a system that cannot assess its semantic decisions will fail to implement safe degradation, human-in-the-loop escalation, or evidence-based verification. While uncertainty quantification alone will not guarantee semantic correctness, it enables agentic systems to recognize when they cannot reliably preserve meaning, make uncertainty-aware decisions, and provide auditable uncertainty provenance alongside decision traces.

4.4 **Re-design Benchmarks for Semantic Quality**. Current benchmarks focus on new tasks and end-to-end scores that are not suitable for diagnosing semantic failures. On the one hand, a new dataset can improve LLMs by providing data for training or fine-tuning LLMs. However, static metrics cannot be used to evaluate semantic quality attributes, such as the degree of drift in meaning, the rate of loss of semantic constraints, and the fraction of evidence-supported generations. Therefore, we need a new benchmark style that compares the semantic quality attributes to test and understand how individual SPL components behave across the multiple phases in the workflows. For example, the semantic contract retention rate could be evaluated by decomposing an existing end-to-end task into intermediate stages and verifying the semantic correctness of the generation at each step. Such benchmarks could be used as a semantic unit test in natural language programming.

4.5 **Open-Source for Semantic Programming**. Open-source projects for AAPL have accelerated the development cycle for agentic AI systems. Many techniques, such as prompt optimization, guardrails, and structured decoding, are now commodities. We need open-source libraries for core functional components, the design of the semantic frames, engines for processing semantic constraints, telemetry for semantic lineage, and a semantic evidence verification module.

## 5. Alternative Views

5.1 **Research should continue as-is**. "*Agentic AI systems need more data on new problem sets that are not exposed to the current LLM pre-training set. By creating new benchmarks, designing new tasks that reveal the weakness of LLMs, and scaling up or specializing model architecture, we can continue to improve the performance.*" While continuing the current direction could yield incremental gains, our study emphasizes that achieving a higher benchmark score does not imply that agentic AI systems achieve higher quality. We observe that the primary barriers to deploying sophisticated agentic

AI systems are not their performance, but stem from system-level quality issues – especially, semantic quality issues. Benchmarks tend to measure when an agent can successfully finish a task under fixed settings, but they don't evaluate system reliability or robustness.

5.2 **Semantic programming is not for research**. "*Semantic programming and semantic quality attributes are not new. They have been studied for a long time in AI research. If we advance the quality of current engineering practice for AFL and AAPL, semantic issues will be resolved.*" Our study indicates that semantic failures do not originate from interface-level bugs or schema-level errors. They are inherent properties of natural language that LLMs also inherit and even amplify in generation. Although individual attributes have been addressed in the literature, these efforts have not been addressed collectively in natural language programming.

5.3 **Agentic AI should prioritize research on other topics**. "*Major progress will come from improvement in reasoning, cognition, or human-level cognitive capabilities.*" The afore-mentioned research directions may improve capabilities of LLMs and agentic AI systems to perform advanced tasks. However, these techniques do not address the root cause of failure modes in agentic AI systems. More advanced cognitive or reasoning systems may exhibit lower-quality behavior in the same way that current agentic AI systems fail. It is also possible that more advanced cognitive capability may amplify the impact of semantic errors.

5.4 **A new programming framework is too much overhead**. "*Building agentic AI on top of a new programming framework introduces too much engineering overhead, and it may introduce biases in research directions.*" We have observed that low-quality implementations fail even before semantic quality is considered – better abstractions tend to reduce overhead in the long run. History supports this view: deep learning research accelerated after automatic differentiation frameworks provided programming abstraction for developing complex mathematical algorithms.

## 6. Conclusion

In this paper, we argue that agentic AI research should prioritize improving the semantic quality of the system. We conducted system analysis and design to identify research gaps from canonical use cases and open challenges proposed in the literature. We show that semantic quality attributes are the root cause of failure modes, and propose a new semantic programming layer as a new research direction for improving the reliability and robustness of the agentic AI systems.

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

# A. Extended Quality Requirements

## A.1. On Semantic Quality

In this section, we present a longer discussion on why the semantic quality in agentic AI systems is not well covered in both software engineering approaches and natural language research.

First of all, the desired level of correctness in agentic AI systems is moving away from assuring quality on its parts or components. The main distinction point is whether the system as a whole can preserve the intended meaning through the multiple stages of LLM inference calls, tool executions, and interaction with multiple agents or other users. The current standard quality model doesn't model the meaning-level contracts that are mediated through natural language because it is not possible to test the meanings in a programmatic way.

Many of the failure modes in agentic AI systems converge to failure for the semantic errors, such as error accumulation over long-horizon LLM inference calls or slight drift in the instruction following behavior, and hallucinating the unknowns. Multi-agent reasoning systems further amplify the semantic quality issues. The difficult-to-catch errors are emerging from the accumulation of delicate changes, and more importantly, the changes in the system configurations, including model versions or hyperparameters, introduce stochasticity that results in non-reproducible outcomes.

**Current Software Quality Models**   Most common quality assurance approaches focus on interfaces and the types or schemas. Guardrails for LLMs also follow a similar idea that ensures the validity of the schema, and reliability and safety of the content generation. Those approaches don't address the following aspects: (1) Preserve constraints across the decomposed stages and executions, (2) Verify evidence-supported claims over multi-step traces, (3) Enforcing precondition, postcondition, and semantic scopes for executions, (4) Ensure the semantic contracts over the stages, going beyond checking at the boundary.

**Current Natural Language Application Research**   Common NLP tasks such as classification, question answering, and summarization evaluate static aspects of generations from input to output with dedicated task metrics. It is also common to extend such an approach in agentic AI benchmarks. Most benchmarks are proposed to cover new use cases with LLM-based baseline agents, showing end-to-end accuracy results. In other words, the correctness of the final outcome is overemphasized.

In agentic AI systems, the task should address the following: (1) Bind entities or slots in the types to the generations and side-effects of the system, (2) propagate constraints through the plans, (3) recover from partial failures. The main focus of semantic quality assurance is the meaning that carries over the stages and agents, and the consequence of the meanings that flow in the workflow.

**Recap on Semantic Quality Requirements**   Table 1 summarizes notable quality concepts around semantic quality in agentic AI systems. We can see that the quality attributes in the traditional quality assurance shift to semantic quality attributes in agentic AI systems. More importantly, those attributes are not well covered in the current software quality models and natural language applications research, calling for the actions that prioritize the research and development for improving the semantic quality of agentic AI systems.

| Shift | Traditional QA | Semantic Quality |
|---|---|---|
| *Reliability* Interface validity to **semantic contracts** | Boundary checks on schemas. A call is correct if inputs/outputs validate. | Acceptance depends on meaning over the flow, semantic postconditions on the state. |
| *Robustness* Single turn metrics to **longitudinal invariants** | Per-turn or single-run pass. No visibility into stability across runs or configurations. | Guard semantic or causal invariance. Bound LLM drifts under seeds, prompts, and model/serving updates. |
| *Observability* Logs to **semantic lineage** | Standard logs or telemetry. Cannot reconstruct the meaning flow. | Semantic lineage over agentic workflow, enabling audit and replay at the meaning level. |

| Shift | Traditional QA) | Semantic Quality |
|---|---|---|
| *Reliability, Robustness* Individual correctness to **collective reliability** | Focus on a single model and correctness. Checking single-generation correctness under a fixed configuration. | Prevent hallucination cascades and non-convergence or agent-debates. Correctness under different system configurations. |
| *Reliability, Observability* Local output to **evidence-supported claims** | Judge answers locally, Provenance is optional, so an unsupported claim could propagate. | Every non-trivial assertion carries verifiable provenance. |

*Table 1.* Semantic Quality Requirements. Shift from traditional quality assurance to semantic quality that preserves the meaning across workflows in agentic AI systems.

## A.2. System Requirements

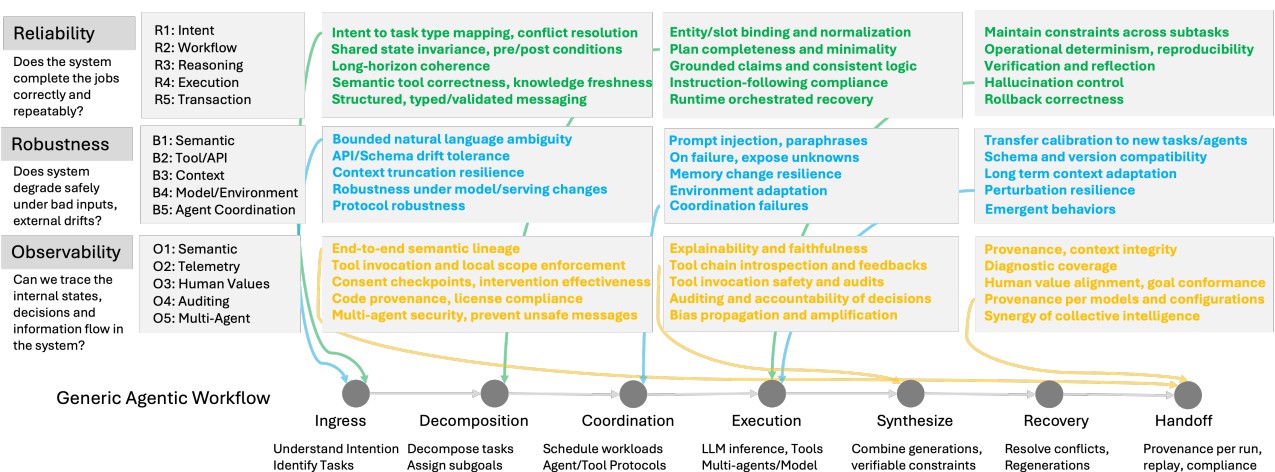

*Figure 3.* Quality Requirements. Under the three major quality criteria, we identified five pillars to organize the quality-related requirements. Some requirements can be mapped to each phase of the generic agentic workflow, and others apply throughout the workflow.

Figure 3 summarizes the quality dimensions, each elaborated into five pillars of requirements. This list is not exhaustive, and the system requirements can be further refined based on target system analysis. Next, we show extended tables on three quality dimensions. This list shows detailed requirements. Note that the following tables are not meant to be an exhaustive list of general agentic AI systems. We present notable quality requirements that are specific to agentic AI systems.

### A.2.1. RELIABILITY DIMENSION

| Pillars | Items | Requirements |
|---|---|---|
| R1 Intent | R1.1 Intent & task-type mapping | Understand the user intention and identify the correct task family |
| | R1.2 Intent conflict resolution | If there exists conflict in priority over policy, user intention, etc., resolve them |
| | R1.3 Entity/slot binding correctness | Preserve correct entity or slot binding, provenance-aware resolution |
| | R1.4 Slot value normalization | The values of semantic types must be normalized |
| | R1.5 Constraint retention | Maintain constraints through decomposition and avoid dropping constraints |

| Pillars | Items | Requirements |
|---|---|---|
| | R1.6 Domain specialization fitness | Meet domain standards (terminology, APIs) and pass domain checks |
| R2 Workflow | R2.1 Plan completeness | Plans include required stages |
| | R2.2 Plan minimality | Plans include only necessary steps |
| | R2.3 Plan–execution adherence | Execution adheres to plan and there's no unplanned generations |
| | R2.4 Operational determinism | Under expected inputs, fixed seeds/configs, etc., outcomes are deterministic. |
| | R2.5 Shared-state invariance | Limit state drift and maintain shared invariants across long horizon workflows |
| | R2.6 Tool affordance semantics | Workflow defined by typed tool models with precondition, postcondition, and the scope |
| R3 Reasoning | R3.1 Long-horizon coherence | Maintain coherent states across multi-step plans. control error accumulation with horizon length |
| | R3.2 Grounded claims | Outputs must be grounded in evidence |
| | R3.3 Verification/reflection effectiveness | Reflection/verification steps should improve the correctness |
| | R3.4 Logical/causal consistency | Produce results that are consistent to logic |
| R4 Execution | R4.1 Knowledge freshness | Keep knowledge up-to-date and ground outputs in authoritative sources |
| | R4.2 Structured call correctness | Tool calls are schema-valid and executable |
| | R4.3 Semantic tool correctness | Schema-valid calls must satisfy semantic postconditions |
| | R4.4 Instruction-following compliance | Outputs adhere to explicit constraints/instructions |
| | R4.5 Hallucination control | Prevent unsupported content generation |
| R5 Transaction | R5.1 Idempotency under retries | Retries must not duplicate side effects in workflows |
| | R5.2 Rollback correctness | Partial failures converge to a safe state through rollbacks |
| | R5.3 Structured messages | Prefer strongly typed, schema-validated messages |
| | R5.4 Runtime-orchestrated recovery | Detect failures at run-time and orchestrate recovery by retries, fallbacks, etc |

*Table 2.* Extended Reliability Dimension Requirements Table. Given the five pillars, we show a list of requirement items relevant to agentic AI systems.

A.2.2. ROBUSTNESS DIMENSION

| Pillars | Items | Requirements |
|---|---|---|
| B1 Semantic | B1.1 Natural language ambiguity | Bound ambiguity when NL ambiguity is unavoidable in prompts/messages |
| | B1.2 Prompt injection robustness | Resist undesired instructions in user content, retrieved docs, or tool outputs |
| | B1.3 Transfer Calibration | Calibrate how task-specific evidence transfers to new tasks |

| Pillars | Items | Requirements |
|---|---|---|
| | B1.4 Paraphrase and causal invariance | Insensitive to the changes that are semantically or causally invariant |
| B2 Tool/API Drift | B2.1 API/schema drift tolerance | Handle tool schema drift without catastrophic workflow failure |
| | B2.2 Fail-closed under outages | Under failed execution, never fabricate tool results. Expose unknown explicitly |
| | B2.3 Schema compliance | Enforce schema validity at boundaries of workflows and maintain backward/forward compatibility |
| | B2.4 Version compatibility | Support mixed-version operation |
| B3 Context & Memory | B3.1 Context truncation resilience | Degrade gracefully as context truncates. preserve key constraints/goals |
| | B3.2 Memory corruption resistance | Resist corrupted memory entries that unexpectedly change future behavior |
| | B3.3 Long-term adaptation | Maintain persistent, hierarchical, relevant context or memory in long-term |
| B4 Multi-agent coordination | B4.1 Coordination failure containment | Contain deadlocks, loops, and cascading error propagation in multi-agent workflows |
| | B4.2 Protocol robustness | Prefer typed/validated communication protocols |
| | B4.3 Emergent behavior containment | Detect and contain unintended system-level dynamics |
| | B4.4 Role drift detection | Detect changes in the role of the agent |
| B5 Environment & Model | B5.1 Robustness under model updates | Updates must not change the system behavior |
| | B5.2 Continual learning stability | Incorporate new knowledge without catastrophic forgetting and change in the behavior |
| | B5.3 Environmental adaptation | Maintain correctness under environmental change such as parallelization, agent-collaboration, etc |
| | B5.4 Perturbation resilience | Resist and gracefully degrade under prompt variants, hyperparameter changes, etc |

*Table 3.* Extended Robustness Dimension Requirements Table. Given the five pillars, we show a list of requirement items relevant to agentic AI systems.

### A.2.3. OBSERVABILITY DIMENSION

| Pillars | Items | Requirements |
|---|---|---|
| O1 Semantic Traceability | O1.1 End-to-end semantic lineage | Reconstruct workflow failures |
| | O1.2 Explainability and faithfulness | Provide step-aligned, faithful explanations reflecting reasoning and tool usages |
| | O1.3 Context integrity | Deliver relevant, consistent context at the right step with traceable provenance |
| | O1.4 Meaning-aware diffs | Compare traces at the meaning level differences |
| | O1.5 Semantic incident taxonomy | Define dropped-constraint, unsupported-claim, etc as incidents to log |
| O2 Telemetry | O2.1 Toolchain introspection and feedback | Expose structured diagnostics so agents can reason about failures/transformations. |
| | O2.2 Diagnostics coverage | End-to-end traces linking workflow phases for monitoring and root-cause analysis. |

| Pillars | Items | Requirements |
|---|---|---|
| | O2.3  Replayable runs | Persist all semantic context to exactly replay workflow |
| O3 Multi-agent | O3.1  Information flow | Trace and monitor inter-agent information flow to prevent error propagation |
| | O3.2  Bias propagation and amplification | Monitor group-wise outcomes and coordination dynamics that can amplify systematic errors |
| | O3.3  Collective intelligence synergy | Demonstrate measurable collaboration gains |
| O4 Human Values | O4.1  Consent checkpoints & intervention | Risk-tiered human gating and effective human-in-the-loop oversight |
| | O4.2  Operational control | Approval gates, predictive risk scoring, escalation mechanisms preventing unsafe actions |
| | O4.3  Goal conformance | Align behavior with user intent and ethical norms and detect goal drift |
| O5 Auditing &Accountability | O5.1  Licensing compliance | Enforce licensing constraints for audit & reuse. |
| | O5.2  Code provenance | Track origin of generated code |
| | O5.3  Auditing and accountability | Standardized auditing and traceable decision logs |
| | O5.4  Provenance stamp per config | Log model, decoding params, serving stack, tool schema versions, etc |

*Table 4.* Extended Observability Dimension Requirements Table. Given the five pillars, we show a list of requirement items relevant to agentic AI systems.

