# OpenReview forum: "Position: Agentic AI Research Should Prioritize Improving Semantic Quality"
_ICML.cc/2026/Position_Paper_Track — Submitted to ICML 2026 Position Paper Track_

### Official Review · Reviewer_ZY2v · 2026-03-13

**Significance:** 2
**Argument Clarity:** 2
**Rating:** 2
**Confidence:** 4

**Questions:**

None

**Alternative Views Section:**

Yes

**Compliance With Llm Reviewing Policy A Conservative:**

Affirmed.

**Discussion Potential:**

2

**Final Justification:**

The author doesn't adequately resolve my issue; I will maintain my score.

**Paper Summary:**

This paper breaks down AI agent failure modes through detailed case studies. The paper show that semantic quality is actually the root cause of these issues, and suggest a new Semantic Programming Layer as a research direction to make agentic systems more reliable and robust.

**Position:**

Yes

**Position In Title:**

Yes

**Related Work:**

2

**Strengths And Weaknesses:**

## Strength

The paper is well-structured and easy to follow. The way the paper categorizes failure modes through specific case studies is good.  By showing exactly where and why current agents break down in real-world scenarios, it provides a very clear and convincing motivation for their proposed framework.

## weakness:

1. While the paper introduces the SPL architecture and various quality dimensions, it lacks empirical evidence (like an ablation study or a simple pipeline demo), to prove they're actually necessary. Relying solely on case studies feels a bit thin, make the qualitative analysis subjective.

2. The other concern is that these 'semantic contracts' make the system too rigid. If you turn natural language into strict code, you kill the agent's talent for handling edge cases. For instance, in business, a good agent needs the flexibility to make exceptions based on unstructured data like customer value. Strict semantic constraints could lead to 'logically correct but business-wrong' decisions.

**Support:**

2

---

> ### Author Rebuttal · Authors · 2026-03-31
>
> We thank the reviewer for the positive assessment of the paper’s structure, clarity, and the use of concrete case studies to motivate semantic quality as a core challenge in agentic AI. We appreciate the thoughtful concerns and respond below.
>
>
>
> **W1 Empirical evidence**
>
> In the position paper track, the main criteria are: (1) a clear position with clear reasoning and evidence, (2) importance in terms of scope, impact, timeliness, etc, (3) inspires constructive discussion, (4) describes an alternative position.
>
> In this submission, our goal is to identify critical research gaps in agentic AI and shift the research focus to underemphasized areas expected to have high impact. Canonical use-case analysis and literature review based on open challenges in agentic AI systems; the failure modes listed in the paper are supported by earlier research in fragment. We try our best to unify them in a single theme under semantic quality and semantic programming in this position paper.
>
>
>
> **Semantic contract is too rigid**
>
> We thank the reviewer for actively engaging in the constructive discussion on the position of this paper. Turning natural language into strict code is one possible way. In specific domains, that might help overall system quality. Our preferred view is to cast LLM reasoning as natural-language programming (writing a program in natural language, or promoting prompts as a program in natural language). In this position paper, we don’t propose any technical solution to the problem, and we shouldn’t.
>
> To engage the reviewer’s comment, we could formulate one possible discussion from “Strict semantic constraints could lead to 'logically correct but business-wrong' decisions”.
>
> In this sentence, it already mentions “constrains”, “logically/syntactically-correct” but “business/semantically-wrong”. They are the key failure modes identified in one of the use case.
>
> Then, the natural question is how to regulate the contract between LLM agents (we view LLM agents as LLM-initiated methods/functions, and contract as an input/output semantic type contract), or do we even need to regulate it? First of all, checking string or enumeration types doesn’t help. This is what’s typically done at AAPL using structured decoding and Pydantic models.
>
> Let’s assume that we want to regulate it. Then, the concern is how to handle edge cases and to be flexible enough to make exceptions based on “value”. One technical direction would be to let the agent understand the intent and be agnostic to particular values that might trigger edge cases or unexpected behavior. In Figure 2, the Constraint processing component handles semantic invariance, pre-post conditions. One research idea is instead of focusing on a single value, reasoning on how the value will change before and after executing the contract (LLM-initiated functions).
>
> We hope this response clarifies the intent and contributions of the paper as a position paper and resolves the concerns raised.

---

> > ### Author Rebuttal · Reviewer_ZY2v · 2026-04-04
> >
> > The author doesn't adequately resolve my issue; I will maintain my score.

---

### Official Review · Reviewer_mWDV · 2026-03-13

**Significance:** 2
**Argument Clarity:** 3
**Rating:** 3
**Confidence:** 4

**Questions:**

See above strengths and weaknesses.

**Alternative Views Section:**

Yes

**Compliance With Llm Reviewing Policy A Conservative:**

Affirmed.

**Discussion Potential:**

3

**Final Justification:**

The author has resolved my issue; however, considering the overall contributions and perspectives presented in the paper, I am maintaining my score and suggest borderline reject.

**Paper Summary:**

This position paper argues that agentic AI research should prioritize improving semantic quality — ensuring that natural language programs encoding agent logic are semantically correct, not merely syntactically valid. The authors analyze three canonical use cases (enterprise workflow agents, software engineering agents, and multi-agent scientific discovery) to identify failure modes rooted in semantic inconsistencies. They propose five semantic quality attributes — semantic contracts, semantic lineage, evidence-supported claims, longitudinal invariance, and collective reliability — and argue that existing frameworks (AFL and AAPL layers) do not address these.

**Position:**

Yes

**Position In Title:**

Yes

**Related Work:**

3

**Strengths And Weaknesses:**

**Strengths**

- Timely and practically motivated problem statement. The gap between agentic AI benchmark performance and real-world deployment reliability is a genuine and pressing concern. The observation that production systems are constrained to short, human-supervised workflows due to reliability concerns — rather than capability limitations — is well-grounded and resonates with practitioner experience.

- The three-layer architecture provides a useful conceptual lens. The distinction between AFL (runtime infrastructure), AAPL (orchestration middleware), and the proposed SPL (semantic program layer) offers a structured way to map existing frameworks and identify the gap. The claim that current open-source and vendor frameworks converge at AAPL and leave SPL unaddressed is a concrete and defensible observation.

**Weaknesses**

- The core concept of "semantic quality" lacks formal grounding, and the distinction from existing work is insufficiently sharp. The paper repeatedly asserts that semantic quality is distinct from syntactic correctness and cannot be addressed by traditional software engineering, but it does not provide a formal or operational definition that would allow researchers to measure or test for semantic quality in practice. Terms like "meaning-level negotiations" and "preserving semantics across workflows" are evocative but imprecise. Without a clearer formalization, it is difficult to evaluate whether the proposed attributes are necessary, sufficient, or mutually exclusive.

- The use case analysis is illustrative but not empirically grounded.
The three canonical use cases are described as "common failure modes," but the paper provides no empirical evidence that these failure modes are prevalent, frequent, or the primary blockers for deployment. The claim that "primary barriers to deploying sophisticated agentic AI systems stem from semantic quality issues" (Section 5.1) is central to the paper's position, yet it is supported only by assertion and the authors' own framing. Citing concrete deployment studies, incident reports, or at minimum a structured survey of practitioners would substantially strengthen the argumen

**Support:**

2

---

> ### Author Rebuttal · Authors · 2026-03-31
>
> We thank the reviewer for the positive assessment of the timeliness, the usefulness, and the clarity of the three-layer architecture view. We appreciate the thoughtful concerns and respond below.
>
> **W1 Semantic Quality**
>
> This concern overlaps with Reviewer FXGp’s W1/Q1 regarding the definition of semantic quality. We therefore refer the reviewer to our answer to FXGp on W1/Q1.
>
> **W2 Empirical Grounding**
>
> There are research/survey papers on deployment studies or incident/failure modes in agentic AI systems. In the introduction and the beginning of Section 3, we cited papers, but didn’t provide details. The observation that the primary barriers to deploying agentic AI systems stem from system-level quality issues (reliability, traceability, etc) is supported by multiple papers. Here, we provide additional details on the literature.
>
> (Pan et al. 2025, Measuring Agents in Production):
> An extensive study on the current limitations of agentic AI in deployment. This paper studies actual usages in production and the current status (human-in-the-loop, short length/pre-defined workflows, mostly using LLMs as-is/no tuning) and points out that reliability is a cause (Finding 13), among others. The current deployment is severely constraining the autonomy, and points out reliability through system-level design, correctness verification/observability, as one direction.
>
> (Mert Cemri, et al 2025, Why Do Multi-agent LLM Systems Fail): An extensive study on the failure modes in multi-agent LLM systems. This paper identifies issues such as system design issues, inter-agent misalignment, and task verification. We cite this as evidence that failures are often caused by coordination and specification problems at the system level, and it corresponds to Semantic Contracts and Collective Reliability in our position paper.
>
> (Wang et al. 2025 AI Agentic Programming: A Survey of Techniques, Challenges, and Opportunities): Focusing on coding agents. This survey paper identifies many of the quality issues. Especially in Section 5/Figure 8, this paper mentions the need for "semantic-aware routing" and the ability to "track state across toolchains" (traceability and observability). This paper also calls for "richer intermediate representations and structured feedback".
>
> (Luo et al 2025, Large Language Model Agent: A Survey on Methodology, Applications, and Challenges) In Section 6, discussing challenges and future trends, this paper points out that maintaining coherence across multi-turn dialogues and the longitudinal accumulation of knowledge requires effective mechanisms.
>
> (Zhu et al. 2025, Where LLM Agents Fail and How They Can Learn From Failures) Introduces error taxonomy, demonstrating how a single "root-cause error" in planning or tool-use propagates through memory and reflection modules, leading to a system failure. They report that current agents are highly vulnerable to cascading failures. This connects to our concept of Longitudinal Invariance and Semantic Drift.
>
> Although the related papers report or discuss general system quality issues in agentic AI systems, and they don’t use the same conceptual taxonomy in this position paper, we see that many of the important quality issues can be consolidated as quality issues applied to the semantic levels. It is our contribution to consolidate the scattered quality issues related to semantics and to provide a functional analysis without technical design or methods.

---

> > ### Author Rebuttal · Reviewer_mWDV · 2026-04-03
> >
> > The author has resolved my issue; however, considering the overall contributions and perspectives presented in the paper, I am maintaining my score.

---

### Official Review · Reviewer_6yTp · 2026-03-13

**Significance:** 2
**Argument Clarity:** 2
**Rating:** 3
**Confidence:** 4

**Questions:**

1.The paper proposes SPL as a distinct layer, but its boundary with advanced orchestration, verification, and guardrail-style middleware is still somewhat ambiguous. Could the authors clarify what would count as a genuinely SPL-specific capability rather than an extension of AAPL?

2.The paper critiques benchmark-driven research, but later also argues for redesigning benchmarks around semantic quality. Is the core claim that benchmarks are the wrong research driver, or that current benchmarks measure the wrong properties?

3.The paper motivates semantic quality with compelling examples, but the evidence is largely illustrative. What kind of empirical study would the authors regard as the strongest validation of their central claim that semantic failures are the main deployment bottleneck for sophisticated agents?

4.The taxonomy of five semantic quality attributes is useful, but some categories seem tightly coupled in practice. Do the authors view these five attributes as truly parallel dimensions, or is there a hierarchy in which some are more foundational than others?

5.The paper emphasizes semantic contracts. Do the authors envision these contracts ultimately being represented as executable constraints, type-like abstractions, ontologies, or still primarily as natural-language specifications? What would make such contracts operational rather than descriptive?

6.The notion of semantic lineage is presented as essential for observability. How does semantic lineage differ fundamentally from existing tracing, provenance, and logging systems, and what new capabilities are required to move from syntactic traceability to semantic traceability?

7.For longitudinal invariance, how should one distinguish unacceptable semantic drift from legitimate diversity in outputs, especially in open-ended tasks where multiple plans or answers may all be valid?

8.For collective reliability, does the paper attribute failures primarily to individual agents, coordination protocols, or the broader topology of the multi-agent system? A more explicit decomposition might help clarify where intervention should occur.

9.The paper calls for uncertainty quantification for semantic decision-making. How do the authors see semantic uncertainty differing from more standard notions of predictive uncertainty or calibration error, and what might be the right object to quantify?

10.The paper calls for integrating four SPL-oriented components: (1) semantic knowledge representation, (2) constraint processing, (3) semantic quality management, and (4) domain-specific rules. Could the authors elaborate more concretely on how these components should be incorporated into existing agent workflows and programming frameworks, and what they see as the main technical bottleneck in unifying them into a coherent, practical system?

**Alternative Views Section:**

Yes

**Compliance With Llm Reviewing Policy A Conservative:**

Affirmed.

**Discussion Potential:**

2

**Ethics Review Area:**

["Other Expertise"]

**Final Justification:**

Partially resolved

**Paper Summary:**

This position paper argues that agentic AI research should prioritize semantic quality over continued emphasis on benchmark-centered task completion. The paper claims that many deployment failures arise not from interface bugs or schema violations, but from meaning-level errors that accumulate across multi-step workflows. To frame this problem, the authors identify five semantic quality attributes: semantic contracts, semantic lineage, evidence-supported claims, longitudinal invariance, and collective reliability, and relate them to standard system qualities such as reliability, robustness, and observability. The paper further proposes a three-layer architecture consisting of AFL, AAPL, and a new Semantic Program Layer, arguing that current research is concentrated in the lower layers while the main unmet need is semantic programming. The paper concludes with a call for SPL-focused frameworks, uncertainty quantification, benchmark redesign, and open-source infrastructure for semantic programming.

**Position:**

Yes

**Position In Title:**

Yes

**Related Work:**

2

**Strengths And Weaknesses:**

Strengths

1.The paper presents a clear and distinctive position, and it supports that position with concrete failure scenarios. A major strength is that the authors do not merely state that semantics matter; they argue that semantic failures are a primary barrier to deployment and illustrate this claim with representative failure modes such as semantically wrong but syntactically valid outputs, loss of constraints across workflows, and breakdowns in multi-agent coordination.

2.The topic is highly relevant to practical agent deployment. The paper focuses on a real and timely issue: agents can perform well on end-to-end benchmarks yet still fail in realistic settings because meaning is not preserved across steps, tools, or interacting components. This is a meaningful concern for the ICML community, especially given the current interest in agent systems, tool use, and reliable deployment.

3.The five semantic quality attributes form a relatively complete conceptual framework. The taxonomy of semantic contracts, semantic lineage, evidence-supported claims, longitudinal invariance, and collective reliability gives the paper structure and helps organize the discussion beyond a generic call for “more reliable agents”.

4.Mapping semantic quality to classical system quality dimensions is a good design choice. The paper does not detach itself from established systems thinking. Instead, it connects semantic quality to reliability, robustness, and observability, which makes the position easier to interpret within a broader software and systems context.

5.The proposed SPL layer is conceptually stimulating. The three-layer architecture gives readers a concrete way to understand the authors’ claim that current research is concentrated in runtime infrastructure and orchestration, while the deeper challenge lies in writing semantically correct natural-language program logic.

6.The action-oriented discussion at the end is relatively practical for a position paper. The paper goes beyond diagnosis by proposing research directions around SPL frameworks, uncertainty quantification, benchmark redesign, and open-source components, which makes the paper more constructive and discussion-worthy.

7.The cited literature appears relevant and up to date with the stated theme. The paper positions itself alongside work on agent frameworks, workflow orchestration, formal methods, semantic web technology, natural language programming, and emerging agent tooling, which suggests that the authors are engaging with the right research context.

Weaknesses

1.The SPL proposal is interesting, but its boundary with AAPL remains somewhat unclear. The paper argues that AAPL focuses on orchestration and scalability while SPL centers semantic correctness, yet many SPL components such as constraint processing, DSLs, ontology, and logic patterns appear adjacent to advanced orchestration, guardrails, or verification-style middleware. The distinction is promising, but not yet fully sharpened.

2.The critique of benchmark-driven research has merit, but it is occasionally stated too strongly. The paper is right that static end-to-end scores do not adequately diagnose semantic failures, but benchmarks themselves could be redesigned to target semantic quality. In that sense, the paper’s criticism may be better framed as a critique of current benchmark design rather than of benchmark-centered evaluation altogether.

3.The failure analysis is persuasive, but the evidence remains mostly illustrative rather than systematic. The scenarios and examples are plausible and well chosen, but the paper does not provide large-scale deployment evidence, controlled empirical analysis, or a systematic case-study methodology. As a result, the argument is compelling at the conceptual level but less established empirically.

4.Some of the five semantic quality attributes appear to overlap. In practice, semantic contracts, semantic lineage, and evidence-supported claims may be tightly coupled, while longitudinal invariance and collective reliability may read more like system-level manifestations than attributes at exactly the same level of abstraction. The taxonomy is useful, but its internal layering could be clarified further.

**Support:**

2

---

> ### Author Rebuttal · Authors · 2026-03-31
>
> We thank the reviewer for the careful reading and thoughtful assessment. It was our pleasure to answer the questions. We address the main concerns and questions below.
>
> **Q1 SPL and AAPL Boundary**
>
> AAPL addresses how to integrate LLM stochastic behavior into regular programs (orchestration, guardrails) and how to handle natural language prompts like regular programs (verification). The genuine SPL-specific capability addresses the question of how to write high-quality natural language programs.
>
> **Q2 Benchmarks**
>
> Our primary concern around end-to-end benchmarks stems from our experience of observing that correct end results come with incorrect (human interpretation, semantics) process, and incorrect final answer delivered by a partially correct process. We would like to rephrase our position to avoid the impression that the benchmark is wrong driver.
>
> **Q3 Evidence**
>
> This question overlaps with Reviewer mWDV’s W2 Empirical Grounding.  We therefore refer the reviewer to our answer to mWDV.
>
> **Q4 Five quality attributes**
>
> Those quality attributes (semantic contracts, invariance, lineage, collective-reliability, evidence-supports) spread over 3 quality dimensions (reliability, robustness, observability).
>
> These attributes are the artifact of system requirement analysis. The semantic contract column is a collection of non-functional requirements. They are the refinement from a larger set of requirements analysis in the appendix that includes items beyond semantic quality.
>
> Since they are a grouping of non-functional requirements, we admit that some items could be placed  differently. But each column has its counterpart in the traditional software quality attributes. Regarding more fundamental attributes, we see that the semantic contract is elementary to all others.
>
> **Q5 Semantic contract**
>
> From our experience, prompts can be managed mostly by Pydantic model or a frame in AI with natural language commands. The data specifications can be embedded in the frame declaration. Enforcement of the semantic contract could be directly applied to the ontology of slots.
>
> **Q6 Semantic lineage**
>
> This is an extension of the existing data lineage concept. The new capabilities are lifting existing logging and tracing to the level of checking the meanings. The lifting toward semantic lineage would allow answering why some agentic AI workflows failed to generate the desired output, possibly due to issues in the underlying semantic model.
>
> **Q7 Longitudinal invariance**
>
> The unacceptable drift is related to the erroneous behavior of LLMs, where a minor change on surface form results in radically different outcomes. We often observe this semantic drift issue when the LLM workload is underspecified and ambiguous. The diversity of the generation of the agentic pipeline could be distinguished from unacceptable drift if the whole generation over the pipeline retains semantic invariance.
>
>
>
> **Q8 Collective reliablility**
>
> Coordination issues or topology issues are well addressed in the literature. The main failure we considered in the position paper is the incompatibility of the semantic spaces of different agents (e.g., different LLMs at the same scale or across scales). We separated this from longitudinal invariance because the quality issues stem from interactions among LLM agents.
>
> **Q9 Uncertainty quantification**
>
> The intention of prioritizing UQ in this position paper is to highlight its importance in decision-making about the generation of LLMs.
>
> We think the standard principle remains the same, but the question is what scoring function (or calibration features) is suitable for the target system. Due to the complexity of the agentic AI pipeline, we need more advanced methods.
>
> **Q10 SPL-oriented components**
>
> We assign functional requirements into four components in the functional architecture. Each component holds consolidated functional features. The question involves design issues in agentic AI workflows. We present one possible idea.
>
> * Knowledge representation: declaration of agent profile in the A2A protocol is standard. We could lift this agent specification to create a semantic contract between agents. Each agent (LLM) exposes their semantic data models and creates an interaction when those semantic data models are compatible.
>
> * Semantic constraint processing: Reviewer ZY2v raised similar questions regarding “the semantic contract is too rigid”. Therefore, we refer to the answer to ZY2v.
>
> * Semantic quality management: The policy to ensure a high semantic quality pipeline can be incorporated into the current workflow. One direction is to make evidence of generation and its verification mandatory.
>
> * Domain-specific rules:  The business logic has its own language that differs from the generic ontology model. Such domain-specific requirements can be explicit in the declaration of the semantic model to be used with LLMs.

---

> > ### Author Rebuttal · Reviewer_6yTp · 2026-04-04
> >
> > The author has resolved some of my concerns. However, given the paper's overall contributions and the reviewers' comments, I will lower my score.

---

### Official Review · Reviewer_FXGp · 2026-03-19

**Significance:** 1
**Argument Clarity:** 2
**Rating:** 2
**Confidence:** 4

**Questions:**

1. How do authors formally define "semantic quality"? Can they propose some specific, quantifiable metrics that distinguish it from existing concepts such as robustness or faithfulness? We expect a good-quality position paper built on explicit math foundations instead of vague concepts.

2. Can the authors provide empirical evidence (even small-scale experiments on several representative datasets) demonstrating that improving the proposed attributes leads to better system performance?

3. I'm also curious how the proposed SPL differs in practice from existing approaches such as program-of-thoughts and tool-augmented reasoning? More discussions or elaborations here are expected.

4. As we know, the semantic quality is a broad concept. So what are the most tractable sub-problems/attributes/dimensions that the community should prioritize first?

**Alternative Views Section:**

Yes

**Compliance With Llm Reviewing Policy A Conservative:**

Affirmed.

**Discussion Potential:**

2

**Final Justification:**

After reading authors' feedback, I believe that my concerns have not been addressed. Moreover, adequately resolving these issues would require substantial revisions that go beyond the scope of the original submission. Therefore, I do not believe the paper is suitable for acceptance in its current form. I will maintain my original scores.

**Paper Summary:**

This position paper intends to study an important concept in agentic AI systems: the gap between task completion benchmarks and system-level quality. The paper argues that current research overly focuses on task success metrics while neglecting **semantic quality**, such as the correctness and consistency of meaning across multi-step workflows. The authors introduce five attributes of semantic quality: semantic contracts, semantic lineage, evidence-supported claims, longitudinal invariance, and collective reliability. Then, they analyze failure modes across several representative user cases (enterprise workflows, software engineering agents, and multi-agent reasoning). Based on this analysis, the paper proposes a new **Semantic Program Layer** as an architectural abstraction for improving semantic quality in natural language programs that encode agent logic. The paper concludes with a call to shift research focus toward semantic quality and outlines several high-level future directions.

**Position:**

Yes

**Position In Title:**

Yes

**Related Work:**

3

**Strengths And Weaknesses:**

**Strengths:**
1. This paper mentions an important limitation of current AI agentic research, that is over-reliance on task completion metrics while ignoring quality dimensions, which leads to unreliable and brittle workflows. This is a timely and impactful topic in recent years' agent trend.

2. In the paper, the authors employ multiple real-world examples like enterprise workflow agents, software engieering agents, and multi-agent reasoning for scientific discovery, to illustrate the agent failures among proposed quality dimensions including reliability, robustness, and observability.

3. The authors proposed a three-layer architecture of AFL/AAPL/SPL in the Figure 2. This is an insightful abstraction for identifying current research gaps:  it is essential for the community to pursue research and development effort to ensure semantic quality of agentic AI systems by building SPL on top.

**Weaknesses:**

1. The notion of “semantic quality” in this paper is broad and lacks exact definitions. The five attributes (semantic contracts, semantic
lineage, evidence-supported claims, longitudinal invariance, and collective reliability) overlap conceptually without clear mathematical formalization or measurable criteria.

2. Overall, the paper is conceptual and does not provide experiments, or any quantitative evaluation to support its claims. The proposed position lacks grounded evidence.

3. While the paper advocates prioritizing semantic quality, it does not explicitly discuss which specific research questions or methods should replace or additionally enhance current approaches. In fact, it's still unclear if such semantic quality metrics may conflict with the previous task completion metrics. I mean, even though some approaches can somehow enhance the agent workflows semantic quality, they may also reduce the task completion rates simultaneously, which is definitely unexpected in the real-world deployment.

4. As far as I know, many discussed issues (e.g.,  reliability, robustness, observability) are already well-known in previous LLM and agent research papers. However, this paper does not persuasively show how its framework promotes beyond existing perspectives.

**Support:**

1

---

> ### Author Rebuttal · Authors · 2026-03-31
>
> We thank the reviewer for the careful reading and for recognizing the importance and timeliness of addressing the gap between task-completion benchmarks and system-level quality in agentic AI. We appreciate the detailed feedback and respond below by clarifying the intent, scope, and contributions of this position paper
>
> **W4 Many discussed issues are known**
> In the AI/ML literature, quality metrics are not well-defined. For example, robustness and reliability are distinct pillars in the quality models of ISO/IEC 25010, but they are often used interchangeably in the literature. We can always find papers that use those two terms in the opposite way. Our position starts with recognizing the importance of quality models, and in this position paper, we narrow them to a smaller subset: reliability, robustness, and observability. They are well-known and established concepts. We excluded security and privacy because they are well addressed in the recent study.
>
> **W1/Q1 Semantic Quality Definition**
> When it comes to agentic AI, essentially, software systems that integrate LLMs as a component for intelligence (because in the absence of LLMs, we wouldn’t call it agentic AI nowadays, it could be expert systems in the 80s),  the quality of the system is dictated by the quality of LLMs.
>
> The quality of LLMs can be approached from multiple viewpoints; from a software engineering perspective, how network systems support high-quality LLM service, etc, collectively captured by AFL (better service);  more recent attempts to make LLMs integrated programs like conventional programs, captured by AAPL (more like programs than prompts).
>
> Our reasoning in the use cases and failure mode analysis reflects that programs in LLMs or natural language programs extend toward a new axis, the semantics. For example, the meaning of a variable name in a conventional programming language doesn’t impact the execution of software, but chaning the name of a field in the prompt significantly impacts the overall behavior.
>
> Our **definition** of semantic quality builds on well-known standard definitions of reliability, robustness, and observability. We apply well-defined definitions to the semantic-aware system requirements of agentic AI systems (Figure 1). A concrete metric can be derived in a mechanical manner from the semantic quality requirements in Figure 1. For example, reliability concerns correctness; a reliable semantic contract measures the correctness of the semantic contract. More concretely, we may count how many semantic contracts (input request vs. response) are violated by checking their content, rather than just their type in structured decoding. It is out of the scope of this position paper to determine what good metrics are, and we believe it is the scope of technical research.
>
> **Q3 SPL differs in PoT**
> SPL is a functional layer for writing semantically correct agentic AI programs. Many of them are rooted in existing semantic technology (DSL/ontology) and logic/probabilistic programs (Knowledge representation/Constraints processing) in AI research. This position paper suggests a direction that introduces an additional abstraction layer on top of current agentic AI frameworks, focusing solely on natural language programming aspects in Agentic AI, leveraging rich prior research in AI.
>
> Program-of-thought or tool usage in reasoning is a particular method to improve the correctness of LLM-generated output. Program-of-thought is a type of CoT prompting method that generates thought in a structured, programmatic way, like Python code. If we map the SPL functional architecture to the Program-of-Thought, a particular procedure, we would map it to the DSL and the ontology component, since the Program of Thought decided to use a particular domain-specific language to solve a restricted set of problems (numerical problems with Python scripts). However, we see that it fits better with the AAPL component that enforces contracts by structuring the thought through a sequence of program codes.
>
> **Q4 subproblems** The subproblems can be formulated for each component in Figure 1 or Figure 2.We think the most tractable and impactful sub-problems in this direction are: (1) systematic study on quality metrics that targets concrete applications, (2) natural language programming approach for identifying reusable prompt patterns as a library, (3) new inference techniques that consider constraint processing at the semantic level, (4) alignment of semantic contracts (input/output semantic types) between agents (functionality) in multi-agent system, (5) moniotring semantic drift and nuiance across different agents in multi-agent system to correct sematnic errors, etc.
>
> We hope this response clarifies the intent and contributions of the paper as a position paper and resolves the concerns raised.

---

> > ### Author Rebuttal · Reviewer_FXGp · 2026-04-03
> >
> > Thank you for your response. However, I feel that my concerns have not been sufficiently addressed. Moreover, adequately resolving these issues would require substantial revisions that go beyond the scope of the original submission. Therefore, I do not believe the paper is suitable for acceptance in its current form. I will maintain my original scores.

---

### Decision · Program_Chairs · 2026-04-30

**Decision:**

Reject

**Comment:**

While the reviewers appreciated the timeliness and practical motivation of addressing agentic AI failure modes, they ultimately agreed that the manuscript lacks the foundational rigor necessary. Even within the context of a position paper, the reviewers noted that the core concept of "semantic quality" lacks a formal definition, the five proposed attributes overlap conceptually, and the proposed Semantic Program Layer is not sufficiently distinguished from existing orchestration middleware. The arguments rely heavily on illustrative case studies rather than structured empirical grounding, user surveys, or formal logic, making the position ultimately unpersuasive in its current form.